# The Combined Power of Double Mass Curves and Bias Correction for the Maximisation of the Accuracy of an Ensemble Satellite-Based Precipitation Estimate Product

Nutchanart Sriwongsitanon [1,*], Chanphit Kaprom [1], Kamonpat Tantisuvanichkul [1], Nattakorn Prasertthonggorn [1], Watchara Suiadee [2], Wim G. M. Bastiaanssen [3] and James Alexander Williams [1,4]

1   Remote Sensing Research Centre for Water Resources Management (SENSWAT), Department of Water Resources Engineering, Faculty of Engineering, Kasetsart University, Bangkok 10900, Thailand; chanphit.kap@ku.th (C.K.); kamonpat.t@ku.th (K.T.); nattakorn.pr@ku.th (N.P.); james@grchydro.com.au (J.A.W.)
2   Royal Irrigation Department, 811 Samsen, Nakornchaisri, Dusit, Bangkok 10300, Thailand; watchara04_rid@icloud.com
3   IHE-Delft, Institute for Water Education, Westvest 7, 2611 AX Delft, The Netherlands; wbastiaanssen@hydrosat.com
4   GRC Hydro Pty Ltd., Level 20/66 Goulburn St, Sydney, NSW 2000, Australia
*   Correspondence: fengnns@ku.ac.th; Tel./Fax: +66-2-5791567

**Abstract:** Precise estimation of the spatial and temporal characteristics of rainfall is essential for producing the reliable catchment response needed for proper management of water resources. However, in most parts of the world, gauged rainfall stations are sparsely distributed and fail to properly capture the spatial variability of rainfall. Furthermore, the gauged rainfall data can sometimes be of short length or require validation. Following this, we present a procedure that enhances the trustworthiness of gauged rainfall data and the accuracy of the rainfall estimations of five satellite-based precipitation estimate (SPE) products by validating them using the 1779 gauged rainfall stations across Thailand. The five SPE products considered include CMORPH-BLD; TRMM-3B42; CHIRPS; CHIRPS-PL; and TRMM-3B42RT. Prior to validation, the gauged rainfall dataset was verified using double mass curve (DMC) analysis to eliminate questionable and inconsistent readings. This led to the improvement of the Nash–Sutcliffe Efficiency (NSE) between the station of interest and its surroundings by 13.9% (0.758–0.863), together with an average 11.8% increase with SPE products, whilst dropping only 7% of questionable dataset. Three different bias correction (BC) procedures were applied to correct SPE products using gauge-based gridded rainfall (GGR). Once DMC and BC procedures were implemented together, the performance of the SPE products was found to increase significantly. Finally, the application of the ensemble weighted average of the three best-performing bias-corrected SPE products (Bias-CMORPH-BLD, Bias-TRMM-3B42, and Bias-CHIRPS) further enhanced the NSE to 0.907 and 0.880 in calibration and validation time periods, respectively. The proposed DMC-based correction SPE and the weighting procedure of multiple SPE products allows for an easy means of obtaining daily rainfall in remote locations with sufficient accuracy.

**Keywords:** gauge-based gridded rainfall; satellite based precipitation estimate; double mass curve; bias correction; ensemble product

## 1. Introduction

Accurate spatial distribution of rainfall is essential for producing reliable runoff estimates for water resource management. Interpolation methods such as geostatistical Kriging; Inverse Distance Weighted (IDW); Thiessen Polygon; and Thin Plate Spline (TPS) [1–3] are often used to generate spatial rainfall estimates from the gauged rainfall measurements. However, uncertainties arise if the observation network becomes increasingly scarce. Although there are numerous gauged rainfall stations in Thailand's river basins, their areal

distribution varies from one basin to another, depending on the topography and anthropogenic impacts. For example, the floodplain of the Chao Phraya basin, where Bangkok is situated, has a rain gauge density of 78.6 km$^2$/station. In contrast, the rain gauge density in mountainous areas such as the Salawin river basin is 1469.7 km$^2$/station. Furthermore, the trustworthiness of gauged rainfall data can sometimes be questionable. Factors including the aerodynamic design of the stations and the effects of wind and wetting losses increase the likelihood of systematic errors [4–6].

An alternative/supplement to gauged rainfall measurements is the use of satellite-based precipitation estimate (SPE) products, which provide an easy alternative to ground observations and are effective in spatiotemporally estimating the variability of precipitation at high spatial and temporal resolutions [7]. In addition, the incorporation of multiple earth observation sensors has been shown to further improve the accuracy, coverage, and spatiotemporal resolution of rainfall estimates [8,9]. Among many high-resolution SPE products, the Tropical Rainfall Measuring Mission (TRMM) Multi-satellite Precipitation Analysis (TMPA) and the Climate Prediction Centre Morphing Technique (CMORPH) have been recognised as the two better-performing products [10–13]. Even though TMPA products (Version 6 and Version 7) performed better than CMORPH in most studies [14–17], CMORPH showed better performance in some, including the investigations carried out by Shen et al. [18] and Li et al. [19]. The TMPA Version 7, which is the latest version, exhibits improved accuracy of rainfall estimation in both real-time (3B42RT) and post-real-time (3B42) products over its predecessor (Version 6) [20–23]. Furthermore, Xue et al. [24] and Zulkafli et al. [25] have utilised TRMM-3B42V6 and TRMM-3B42V7 in estimating runoff, and the results showed that Version 7 generated hydrographs closer to the observed hydrographs than those of Version 6.

However, the poor spatial resolution (0.25°, which is equivalent to a spatial coverage of approx. 770 km$^2$/pixel) of the TMPA and CMORPH products is a major drawback in capturing the detailed spatial variations of rainfall required for accurate estimation of the runoff of the basin. Therefore, the Climate Hazards Group Infrared Precipitation with Stations (CHIRPS) version 2.0 was developed by Funk et al. [26] using three main components. These include the Climate Hazards group Precipitation Climatology (CHPclim), the satellite-only Climate Hazards group Infrared Precipitation (CHIRP), and the station blending procedure to produce CHIRPS. These products provide rainfall estimates at the finest spatial resolution of 0.05° (coverage of approx. 30 km$^2$/pixel). Duan et al. [27]; Poortinga et al. [28]; and Simons et al. [29] compared the accuracy of SPE products in Italy, China, and Vietnam using TRMM-3B42V7, CMORPH, and CHIRPS and concluded that CHIRPS was the second-best-performing rainfall product after TRMM-3B42V7.

The accuracy of SPE is usually enhanced by comparing and bias-correcting them with the gauged rainfall data [30]. Among many BC procedures, linear bias correction [31–33], distribution transformation [34,35], and regression analysis [36–38] are widely and successfully utilised in various studies.

As SPEs are validated against the gauged rainfall stations, their accuracy is highly dependent upon the quality of the gauged rainfall data used, which is always subjected to some degree of uncertainty. Searcy and Hardison [39] proposed a double mass curve (DMC) for checking the consistency of hydrologic data by comparing the cumulative data for a single station with the cumulative data from several other stations in the area. The DMC has been continuously utilised for many kinds of hydrological data including rainfall [40–42], runoff, sediment [43–46], and aquifer drawdown [47], as well as rainfall–runoff and runoff–sediment relations [48].

The objective of this study is to enhance the accuracy of five SPE products, consisting of TRMM-3B42, CMORPH-BLD, CHIRPS, and the two near-real-time products, which are TRMM-3B42RT and CHIRPS-PL, covering Thailand's river basins and combine them to form a weighted averaged series. Gauged rainfall data from over 1700 stations were utilised to improve the accuracy of the SPE products by applying three BC procedures, consisting of linear bias correction (LBC), bias correction using distribution transformation, and bias

correction using regression analysis (RABC). Prior to its application, the monthly gauged rainfall data from 2001 to 2015 was validated using the DMC procedure. The usefulness of the DMC to increase the accuracy of areal rainfall estimates and to possibly decrease the number of existing gauged rainfall stations was also investigated. Thereafter, the advantage of applying the DMC to the gauged rainfall data along with the BC was evaluated.

## 2. Materials and Methods

### 2.1. Study Area and Data Description

#### 2.1.1. Study Area

Thailand is located at the centre of peninsular Southeast Asia between 5° N–21° N latitude and 97° E–106° E longitude. Thailand is bordered to the west by Myanmar and the Andaman Sea, to the northeast by Laos, to the southeast by Cambodia, and to the south by the Gulf of Thailand and Malaysia (see Figure 1). The country covers a total area of 515,934.08 km$^2$ and is divided into 25 main river basins with areas varying from 4148 km$^2$ to 70,943 km$^2$. The altitude ranges from −6 m (MSL) to 2565 m (MSL). The northern region is the mountainous area and is the origin of the Ping, Wang, Yom, and Nan River Basins—the main tributaries of the Chao Phraya River Basin, which covers the area of nearly one-third of the country. Thailand has a tropical climate affected by the southwestern and northeastern monsoons, which cause the rainy season (May to October) and the dry season (November to April), respectively, for most of the country. This is except for the southern part, which gains extra rainfall for the first three months of the dry season. The annual average rainfall (2001–2015) of the southern areas is around 2045 mm, while that of other regions is approximately 1402 mm.

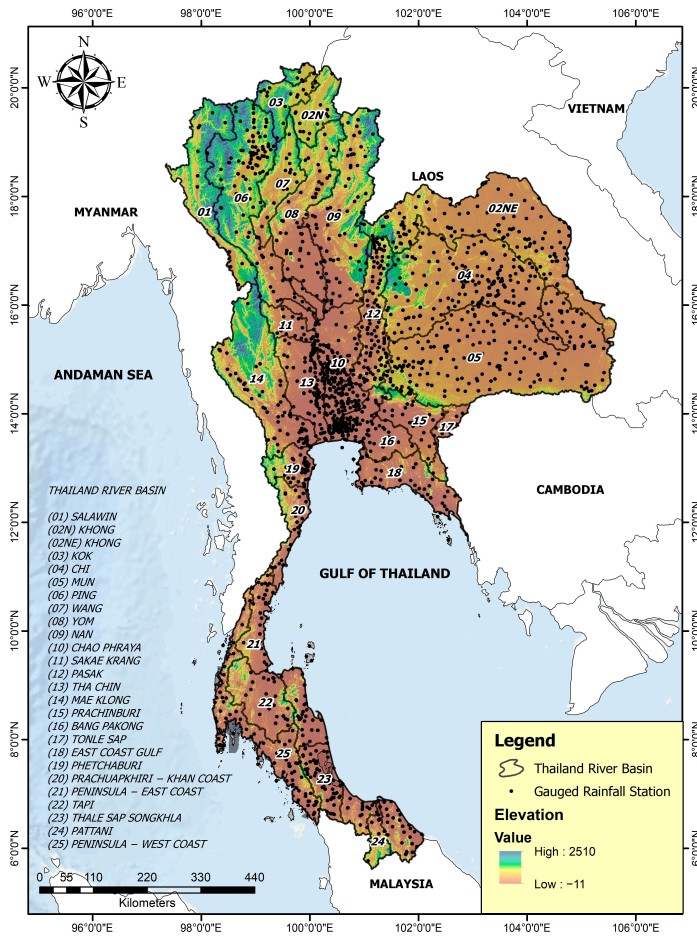

**Figure 1.** Spatial distribution of 1779 gauged rainfall stations used in the study.

### 2.1.2. Gauge Rainfall Data

Monthly rainfall data from 1779 non-automatic gauge rainfall stations during the period from 2001 to 2015 were used in this study. Around 65% of these stations were operated by the Thailand Meteorological Department (TMD), and 33% were operated by the Royal Irrigation Department (RID). The remaining stations were managed by other government agencies. Figure 1 shows these gauged rainfall stations, which were already checked as to whether they were situated at the right positions; then, the incorrect stations were relocated to the appropriate locations. The majority of these stations are densely located in the central region, which is concentrated by the irrigated areas.

### 2.1.3. Satellite-Based Precipitation Estimate (SPE) Products

Five SPE products were selected for this study, comprising CMORPH-BLD, TRMM-3B42, TRMM-3B42RT, CHIRPS-PL, and CHIRPS. The characteristics of these products are presented in Table 1, which describes the temporal and spatial resolutions, temporal and spatial coverages, latency, and data source. The summaries of these products are described below.

(1) The Tropical Rainfall Measuring Mission (TRMM) was launched in November 1997 through a collaboration between the National Aeronautics and Space Administration (NASA) and the Japan Aerospace Exploration Agency (JAXA). TRMM is a low-earth-orbit satellite equipped with Precipitation Radar (PR), TRMM Microwave Imager (TMI), Visible and Infrared Sensor (VIRS), lightning imaging sensor (LIS), and the Earth's Radiant Energy System (CERES) [49]. The TRMM Multi-satellite Precipitation Analysis (TMPA) products are the combination of infrared (IR) data from geostationary satellites and microwave (MW) data from multiple satellites. The TRMM-3B42 V7 3-hourly precipitation products cover the tropical and subtropical regions with a spatial resolution at $0.25 \times 0.25$ grid scale. There are four steps in creating the products. Step 1, the passive microwave field of view from different sources is calibrated and combined using algorithms such as sensor-specific versions of the Goddard Profiling Algorithm (GPROF). Step 2, the IR precipitation estimates are computed using the histogram matching of monthly MW precipitation estimates. Step 3, the MW and IR precipitation estimates are merged, with IR estimates being utilised to fill in the gap where MW estimates are missing. Rain gauge data are finally utilised to rescale and calibrate the merged precipitation estimates. TRMM-3B42RT product is originally evaluated to provide the near-real-time data and is then bias-corrected using monthly gauge rainfall data from the Global Precipitation Climatology Centre (GPCC) to generate the post-real-time data, TRMM-3B42 product [50–52]. The datasets of these two products were downloaded from NASA's Goddard Space Flight Center website (https://disc2.gesdisc.eosdis.nasa.gov/data/, accessed on 1 April 2020) and utilised in this study.

**Table 1.** Descriptions of the satellite-based rainfall estimate (SPE) products utilised in this study.

| SPE Product | Gauged Observation | Temporal Resolution | Spatial Resolution | Temporal Coverage | Spatial Coverage | Latency | Data Source |
|---|---|---|---|---|---|---|---|
| TRMM-3B42 V7 | GPCC | 3 h | 0.25 × 0.25° | 1998–2019 | 50° S–50° N | 2 months | https://disc2.gesdisc.eosdis.nasa.gov/data/TRMM_L3/TRMM_3B42_Daily.7/2015/01/ Accessed on 1 April 2020 |
| TRMM-3B42RT V7 | - | 3 h | 0.25 × 0.25° | 1998–2019 | 50° S–50° N | 8 h | https://disc2.gesdisc.eosdis.nasa.gov/data/TRMM_RT/TRMM_3B42RT_Daily.7/2015/01/ Accessed on 1 April 2020 |
| CHIRPS-PL V2.0 | GTS and Conagua | 2 days | 0.05 × 0.05° | 1981–2015 | 50° S–50° N | 1 week | https://data.chc.ucsb.edu/products/CHIRPS-2.0/prelim/global_monthly/tifs/ Accessed on 15 April 2020 |
| CHIRPS V2.0 | GPCC, GTS, and Conagua | 1 day | 0.05 × 0.05° | 1981–present | 50° S–50° N | 3 weeks | https://data.chc.ucsb.edu/products/CHIRPS-2.0/global_monthly/tifs/ Accessed on 15 April 2020 |
| CMORPH-BLD V1.0 | CPC unified daily gauge analysis, GPCC | 1 day | 0.25 × 0.25° | 1998–present | 60° S–60° N | 2 months | https://ftp.cpc.ncep.noaa.gov/precip/CMORPH_V1.0/BLD/0.25deg-DLY_EOD/GLB/2015/201501/ Accessed on 30 April 2020 |

(2) Climate Hazards Group Infrared Precipitation (CHIRPS) and Climate Hazards Group Infrared Precipitation with stations (CHIRPS) were developed by the University of California Santa Barbara's Climate Hazards Group to support the United States Agency for International Development Famine Early Warning Systems Network (FEWS NET) [26]. Four steps are involved in producing CHIRPS dataset. Firstly, Infrared Precipitation (IRP) pentad rainfall estimates are first generated using local regressions between TRMM 3B42V7 precipitation analysis pentads and cold cloud duration with a uniform threshold of 235 K. Secondly, the temporal component of IRP pentadal is converted to percentage anomalies and multiplied by the spatial component CHPClim pendatal to produce the Climate Hazards Group IR Precipitation (CHIRP)—the unbiased gridded estimate. The adjusted IRP is then combined with gauged rainfall data from Global Telecommunications System (GTS) and Conagua (Mexico) to create a rapid preliminary version (CHIRPS-PL) (2-day latency). Finally, a later final version (CHIRPS) is delivered within the third week of the following month by using extra gauged rainfall observations, mainly from the USA, Central America, South America, and sub-Saharan Africa [53–56]. CHIRPS-PL and CHIRPS were employed in this study and were downloaded from https://data.chc.ucsb.edu/products, accessed on 15 April 2020.

(3) The National Oceanic and Atmospheric Administration (NOAA) Climate Prediction Centre MORPHing method (CMORPH) generates precipitation data by merging passive microwave-based precipitation estimates from multiple low-earth-orbit (LEO) satellites and the infrared data from multiple geostationary satellites [57]. CMORPH uses thermal IR temperatures to create the cloud systems advection vectors (CSAVs) to fill the gaps where temporal and spatial observations of MW-based rain rates are not available. The CSAVs are later applied to propagate MW-based rain rates in forward and backward directions between two successive MW overpasses using linear interpolation to morph the shape and intensity of the propagated rainfall pattern to produce CMORPH-RAW [27,58]. The CMORPH-CRT is produced by adjusting the CMORPH-RAW against the CPC unified daily gauge-based analysis over land and the pentad Global Precipitation Climatology Centre (GPCC) over the ocean using the probability density function bias correction procedure [59]. The CMORPH–CRT is additionally combined with the gauge analysis using the optimal interpolation technique to generate the CMORPH–BLD product [7]. CMORPH-BLD are available at https://ftp.cpc.ncep.noaa.gov/precip/CMORPH_V1.0/BLD/0.25deg-DLY_EOD/GLB/2015/201501/, accessed on 30 April 2020.

### 2.2. Methods

### 2.2.1. Validation of Gauged Rainfall Data Using the DMC Procedure

Inaccuracies in raw gauged rainfall data necessitate a validation procedure. DMC was used to observe correlations between (a) rainfall depths measured at each station and (b) the weighted average rainfall of the surrounding stations [39]. The weighted average rainfall was calculated using the Inverse Distance Square (IDS) procedure, and the number of stations used in the calculation was chosen to optimise the correlation between these variables.

Once DMC indicated a discrepancy, unreliable data were eliminated from the gauged rainfall dataset, thereby improving the correlation of the DMC. However, although the continual removal of unreliable data would gradually enhance the DMC, excessively doing so would affect the integrity of the gauged rainfall dataset. Therefore, to determine the suitable extent of elimination, a set of DMCs was produced from the gauged rainfall dataset which had data eliminated to different extents, beginning from a minimum Nash–Sutcliffe coefficient (NSE) threshold of $NSE < 0.6$ [60]. Whilst increasing the threshold, the correlation between the DMC and SPE products was monitored. It is suspected that there would be an NSE threshold at which the gauged rainfall data would begin to lose its reliability and, thus, display lower correlations with the SPE products.

### 2.2.2. Effect of the Validity of Gauged Rainfall Data on Their Correlation with SPE

Due to the hypothesis that the accuracy of the final SPE product is dependent upon the quality of gauged rainfall data, a procedure was carried out to verify this. Firstly, the correlation between each of the SPE products and the original gauged rainfall dataset was determined. Each SPE product was then compared with the gauged rainfall datasets, which were used to produce the DMCs in Section 2.2.1 in order to observe any changes in their correlations as the NSE threshold increased.

### 2.2.3. Usefulness of DMC

A method was devised to reinforce the effectiveness of using the DMC procedure to enhance the accuracy of gauged rainfall data. In order to create the correlational improvements as provided by DMCs, this could only be achieved by hypothetically densifying the observation network within the study area. Hence, as a means to replicate the outcome from Section 2.2.1, stations were randomly removed from the gauged rainfall dataset to induce an increase in the root mean square error (RMSE) in the areal rainfall estimates relative to utilising the full dataset. The stations were gradually removed from the dataset until the subsequent increase in RMSE was equivalent to the reduction in RMSE as a result of the DMC procedure.

### 2.2.4. Pixel-Based Comparison of SPE Products

To perform BC on the SPE products, the gauged rainfall dataset must be gridded. Hence, gridded gauged rainfall (GGR) datasets were generated using the validated gauged rainfall datasets. Each pixel was calculated by weighting the rainfall depths at surrounding stations using the IDS procedure. No more than ten gauged rainfall stations within a radial area of 25 km$^2$ were included in the calculation. The three nearest stations were selected if no stations were situated within the specified area. Consequently, the GGR data were compared with SPE products to determine the most accurate product for estimating rainfall in this study.

### 2.2.5. Bias Correction Procedure for SPE Products

The mismatch between the SPE and GGR data limits the ability to directly utilise SPE products for rainfall estimation. This can be appreciably mitigated with bias correction (BC). In this study, three common BC methods were chosen to improve the accuracy of SPE products, comprising linear bias correction (LBC), bias correction using regression analysis (RABC), and bias correction using distribution transformation (DTBC). The GGR data were utilised to correct the SPE products on the BC process. To demonstrate the effectiveness of DMC for eliminating unreliable rainfall data prior to producing the GGR data, the BC procedures were also applied to gridded rainfall data generated from uncorrected (original) rainfall data, which shall be simply referred to as GGR$_{Ori}$ hereafter. The theory of each BC procedure is described below.

(1) Linear bias correction (LBC)

The linear bias correction (LBC) method is a simple and widespread procedure to be used for adjusting the systematic error of *SPE* products [31–33]. A bias corrector was calculated at each pixel as the ratio between the summation of the *GGR* and *SPE* rainfall for the overall time series. This factor was later used to adjust the whole time series of *SPE* dataset, as shown in Equation (1).

$$SPE'_{(i)} = SPE_{(i)} \cdot \frac{\sum_{i=1}^{n} GGR_{(i)}}{\sum_{i=1}^{n} SPE_{(i)}} \tag{1}$$

where $SPE'_{(i)}$ is the bias-corrected *SPE* dataset for $i$th month, $SPE_{(i)}$ is the original *SPE* dataset for $i$th month, $GGR_{(i)}$ is the gauged-based gridded rainfall for $i$th month, and $n$ is number of months, respectively.

(2)   Bias correction using regression analysis (RABC)

Nonlinear regression analysis has been demonstrated to be an effective method in reducing the discrepancy between *SPE* product and gauged rainfall data [36–38,61,62]. In this study, the second-order polynomial regression was chosen to correlate the monthly *GGR* and *SPE* datasets of each pixel, as shown in Equation (2). The bias correctors *a* and *b* were computed and employed to adjust the *SPE* rainfall for the entire time series.

$$SPE'_{(i)} = aSPE^2_{(i)} + bSPE_{(i)} \tag{2}$$

(3)   Bias correction using distribution transformation (DTBC)

Distribution transformation bias correction was applied for the correction of Global Climate Model (GCM) datasets [34]. This method was utilised in this study to adjust *SPE* datasets by substituting their standard deviation and mean with those from the *GGR* data, as presented in Equation (3).

$$SPE'_{(i)} = \left( \frac{SPE_{(i)} - SPE_\mu}{SPE_\sigma} \right) \times GGR_\sigma + GGR_\mu \tag{3}$$

where $SPE_\mu$ is the mean of original raw SPE dataset, $SPE_\sigma$ is the standard deviation of original *SPE* dataset, $GGR_\mu$ is the mean of *GGR* dataset, and $GGR_\sigma$ is the standard deviation of *GGR* dataset. $SPE'_{(i)}$ and $SPE_{(i)}$ are the corrected and original *SPE* datasets for time step *i*, respectively.

### 2.2.6. Cross-Validation of Bias Correction Procedures

The temporal cross-validation technique was used to test the effectiveness of BC procedures. To reduce the effect of overfitting in training datasets, the sliding window validation introduced by Kotu and Deshpande [63] was utilised by subdividing the whole dataset (180 months) into ~70% of data (132 months) for calibration and ~30% (48 months) for validation. At the first iteration, the datasets between the 1st and 132nd month were selected for calibration to evaluate the bias correctors of the SPE datasets. These correctors were then applied to correct SPE datasets of the validation period (133rd to 180th month). For the following iterations, the calibration and validation periods were slid to the following months, and this process was repeated until the whole dataset was used (see Figure 2). The accuracy of bias-corrected SPE products relative to the GGR datasets within the validation period was determined by using NSE. The average NSE value from 180 iterations of sliding windows for each of the SPE products was determined to assess the overall performance.

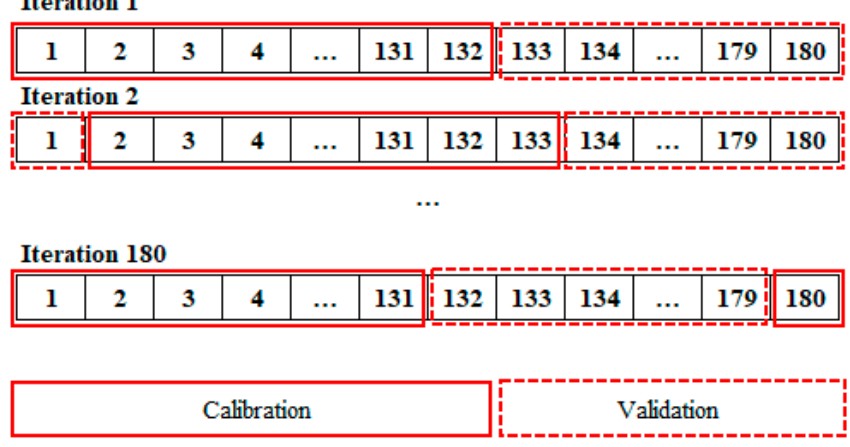

**Figure 2.** The diagram represents the temporal cross validation undertaken over the 180-month dataset. Solid boxes represent the period (70% of total) used for calibration. The remaining months were used for validation, as represented by the dashed boxes.

## 3. Results and Discussion

### 3.1. Effect of Validity of Gauged Rainfall Data on Their Correlation with SPE Products

To create the original DMC, the weighted rainfall depth for each station was calculated using 10 surrounding stations (varying from three to ten due to the uneven distribution of stations). The average NSE between rainfall depths at the stations of interest and the weighted average rainfall depths of the surrounding stations (i.e., the original DMC) was 0.758, as shown in Table 2. The average NSE between the original DMC and each SPE product in decreasing order was CMORPH (0.612), CHIRPS-PL (0.608), TRMM-3B42 (0.575), CHIRPS (0.512), and TRMM-3B42RT (0.354).

The effect of data elimination on the correlation with SPE products is also presented in Table 2. For example, consider the first scenario, wherein 0.51% of gauged rainfall data was removed because they did not reach the NSE threshold of 0.60. Basically, this led to the removal of one gauge rainfall station as the complete time series was found unreliable, leaving us with 1778 stations. This improved the average NSE of the DMC to 0.783. The DMC also showed improved NSE with all SPE products (e.g., for TRMM-3B42, NSE improved from 0.575 to 0.603). By gradually increasing the NSE threshold and dropping the data/stations, it was revealed that the NSE of gauged rainfall data and SPE began to decline at the threshold of 0.84. Meanwhile, this occurred for CHIRPS-PL and CMORPH-BLD at the threshold of 0.95. However, at this threshold the gauged rainfall dataset was significantly compromised due to a 27% removal of data. For this reason, the suitable NSE threshold used to produce the validated gauged rainfall dataset was fixed at 0.84, at which 6.95% of the data were removed, lowering the total number of stations to 1743. This improved the NSE of the DMC by 13.9% from 0.758 to 0.863. The NSE between the gauged rainfall dataset with the SPE products also increased by an average of 11.8%—with CMORPH-BLD, CHIRPS-PL, TRMM-3B42, CHIRPS, and TRMM-3B42RT increasing by 8.3%, 7.1%, 8.7%, 12.9%, and 22.0%, respectively.

This demonstrates the ability of DMC to validate gauged rainfall data, which is important for ensuring that the dataset is reliable for performing BC on SPE rainfall estimates. Having determined the optimal NSE threshold of 0.84, the NSE of the validated gauged rainfall dataset and SPE products was further examined. For each product, the NSE of the monthly rainfall estimates and the corresponding gauged rainfall depth was determined. A cumulative distribution function was produced to show the percentage of gauged and SPE rainfall depth pairs which show the NSE above a certain value, as shown in Figure 3. In total, 34.8% and 27.2% of gauged rainfall depths showed an NSE > 0.8 with CMORPH-BLD and TRMM-3B42, respectively. Moreover, 73.1% of gauged rainfall depths showed an NSE > 0.7 with CMORPH, followed by with CHIRPS-PL (66.0%) and TRMM-3B42 (64.1%).

Further observations can be made with the bracketed values in Figure 3, which show the percentage increase in SPE after validating the gauged rainfall dataset. For instance, for CMORPH-BLD, the percentage of rainfall depth pairs with NSE > 0.8 increased by 16.3% (from 18.5% to 34.8%). The steep slope of the cumulative distribution curves for most products indicates the effectiveness of eliminating unreliable gauged rainfall data in improving their correlation with SPE products. On average, the major improvement was seen for NSE thresholds of >0.7 and >0.8.

**Table 2.** Correlations of the DMC and the SPE products after performing data elimination to different NSE thresholds.

| NSE Threshold | N | Discarded Data (%) | Double Mass Curve | | TRMM-3B42 | | TRMM-3B43RT | | CHIRPS | | CHIRPS-PL | | CMORPH-BLD | |
|---|---|---|---|---|---|---|---|---|---|---|---|---|---|---|
| | | | NSE (Original) | NSE (After) | NSE (Original) | NSE (After) | NSE (Original) | NSE (After) | NSE (Original) | NSE (After) | NSE (Original) | NSE (After) | NSE (Original) | NSE (After) |
| Original | 1779 | 0.00 | 0.758 | - | 0.575 | - | 0.354 | - | 0.512 | - | 0.608 | - | 0.612 | - |
| 0.60 | 1778 | 0.51 | 0.758 | 0.783 | 0.575 | 0.603 | 0.355 | 0.421 | 0.513 | 0.565 | 0.608 | 0.624 | 0.612 | 0.640 |
| 0.65 | 1776 | 0.85 | 0.758 | 0.790 | 0.576 | 0.604 | 0.355 | 0.421 | 0.513 | 0.566 | 0.608 | 0.628 | 0.613 | 0.643 |
| 0.70 | 1774 | 1.38 | 0.758 | 0.801 | 0.576 | 0.610 | 0.356 | 0.425 | 0.514 | 0.569 | 0.609 | 0.630 | 0.613 | 0.647 |
| 0.75 | 1764 | 2.39 | 0.759 | 0.817 | 0.581 | 0.616 | 0.361 | 0.427 | 0.519 | 0.569 | 0.612 | 0.636 | 0.618 | 0.652 |
| 0.80 | 1755 | 4.11 | 0.760 | 0.839 | 0.584 | 0.621 | 0.364 | 0.431 | 0.521 | 0.580 | 0.612 | 0.642 | 0.619 | 0.662 |
| 0.81 | 1752 | 4.96 | 0.761 | 0.845 | 0.584 | 0.621 | 0.364 | 0.429 | 0.522 | 0.579 | 0.613 | 0.644 | 0.620 | 0.662 |
| 0.82 | 1751 | 5.51 | 0.761 | 0.851 | 0.585 | 0.620 | 0.364 | 0.432 | 0.522 | 0.577 | 0.613 | 0.645 | 0.620 | 0.665 |
| 0.83 | 1746 | 6.21 | 0.761 | 0.857 | 0.586 | 0.622 | **0.365** | **0.433** | 0.523 | 0.577 | 0.613 | 0.648 | 0.621 | 0.662 |
| 0.84 | 1743 | 6.95 | 0.761 | 0.863 | **0.586** | **0.625** | 0.365 | 0.432 | **0.524** | **0.578** | 0.614 | 0.651 | 0.622 | 0.663 |
| 0.85 | 1733 | 7.29 | 0.761 | 0.869 | 0.587 | 0.624 | 0.366 | 0.430 | 0.525 | 0.576 | 0.615 | 0.653 | 0.623 | 0.664 |
| 0.90 | 1699 | 13.76 | 0.761 | 0.909 | 0.590 | 0.617 | 0.371 | 0.417 | 0.529 | 0.577 | 0.616 | 0.663 | 0.626 | 0.668 |
| 0.95 | 1605 | 27.21 | 0.763 | 0.953 | 0.597 | 0.616 | 0.378 | 0.412 | 0.539 | 0.572 | **0.619** | **0.666** | **0.632** | **0.673** |
| 1 | 432 | 80.74 | 0.777 | 1.000 | 0.557 | 0.479 | 0.326 | 0.482 | 0.477 | 0.451 | 0.578 | 0.467 | 0.591 | 0.538 |

Note: N is the number of gauged rainfall stations; rows with bolded numbers correspond to the NSE threshold which yields the maximum NSE after dropping the data/stations for each SPE product; the highlighted row is the suitable NSE threshold used to produce the validated gauged rainfall dataset.

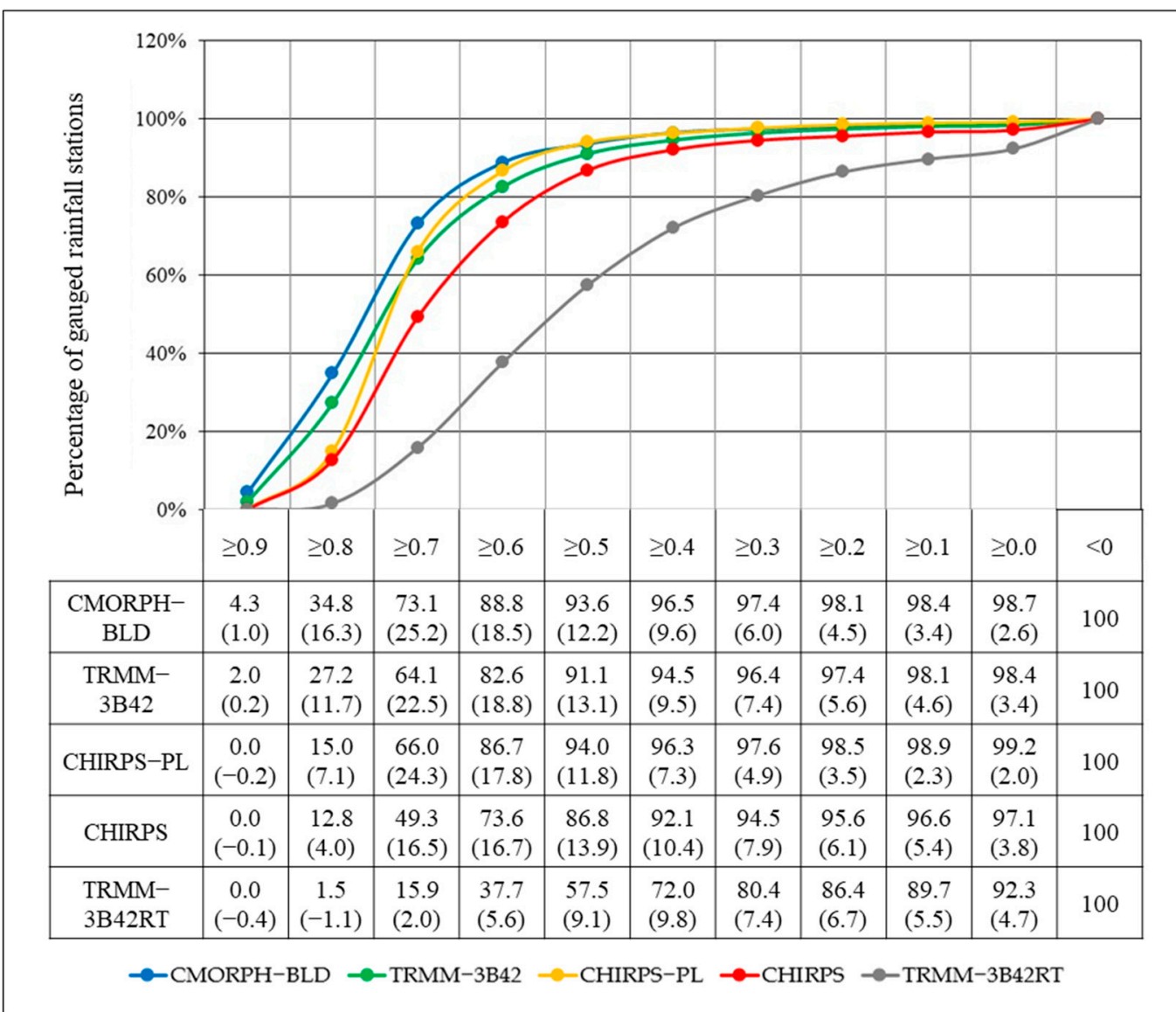

**Figure 3.** Cumulative percentage of gauged rainfall station data which correlate with SPE above certain thresholds.

### 3.2. Usefulness of DMC

The usefulness of the DMC procedure can be further evaluated using Table 3. After the validation of the gauged rainfall dataset with a threshold NSE of 0.84, at which 6.95% of data were removed, the RMSE of areal rainfall estimates was reduced by 16.3 mm. In contrast, the attempt to induce an increase in the RMSE of areal rainfall estimates by 16.3 mm required the random removal of 514 stations from the total of 1743 stations from the validated gauged rainfall dataset. Thus, this suggests that in order to achieve the equivalent NSE improvements between gauged rainfall data and SPE products, the observation network must be significantly densified from 591.7 km$^2$/station to 395.7 km$^2$/station.

**Table 3.** Comparison of the effect of the DMC procedure and the random removal of gauged rainfall stations on reducing and inducing error in the areal rainfall estimates across 25 sub-basins.

| Region | Code | Basin | Number of Station | DMC | | | Random Removal | | | |
|---|---|---|---|---|---|---|---|---|---|---|
| | | | | Discard (%) | Density (km²/St.) | Reducing RMSE (mm) | St. | Remove (%) | Density (km²/St.) | Increasing RMSE (mm) |
| North | 06 | Ping | 101 | 4.9 | 331.7 | 10.8 | 11 | 10.9 | 367.0 | 10.6 |
| | 02N | Khong | 22 | 2.7 | 477.8 | 13.7 | 3 | 13.6 | 528.1 | 12.9 |
| | 08 | Yom | 49 | 5.9 | 479.0 | 16.1 | 16 | 32.7 | 684.2 | 16.9 |
| | 07 | Wang | 21 | 4.0 | 539.7 | 11.3 | 5 | 23.8 | 674.6 | 10.7 |
| | 09 | Nan | 63 | 4.5 | 545.4 | 10.8 | 13 | 20.6 | 671.3 | 10.6 |
| | 03 | Kok | 14 | 5.8 | 561.5 | 14.6 | 3 | 21.4 | 663.6 | 12.0 |
| | 01 | Salawin | 13 | 5.0 | 1592.2 | 23.0 | 4 | 30.8 | 2122.9 | 24.8 |
| Central | 10 | Chao Phraya | 254 | 4.9 | 78.9 | 10.1 | 47 | 18.5 | 96.0 | 10.1 |
| | 12 | Pasak | 120 | 10.4 | 129.1 | 28.1 | 93 | 77.5 | 538.7 | 28.2 |
| | 13 | Tha Chin | 77 | 5.5 | 173.0 | 4.4 | 10 | 13.0 | 195.5 | 4.5 |
| | 11 | Sakae Krang | 18 | 3.8 | 297.4 | 10.8 | 5 | 27.8 | 388.9 | 9.6 |
| North-East | 04 | Chi | 163 | 5.7 | 299.6 | 19.5 | 70 | 42.9 | 517.2 | 19.6 |
| | 05 | Mun | 190 | 5.1 | 370.2 | 15.6 | 59 | 31.1 | 530.4 | 15.5 |
| | 02NE | Khong | 123 | 3.6 | 383.4 | 20.1 | 38 | 30.9 | 548.3 | 20.6 |
| East | 16 | Bang Pakong | 51 | 7.2 | 209.8 | 12.4 | 15 | 29.4 | 289.2 | 12.9 |
| | 18 | East Coast Gulf | 46 | 8.3 | 284.6 | 22.0 | 11 | 23.9 | 363.7 | 20.9 |
| | 15 | Phachinburi | 27 | 5.8 | 372.0 | 20.7 | 10 | 37.0 | 568.9 | 21.6 |
| | 17 | Tonle sap | 8 | 13.8 | 583.7 | 35.0 | 6 | 75.0 | 2043.0 | 37.2 |
| West | 19 | Phetchaburi | 35 | 4.4 | 178.9 | 3.4 | 5 | 14.3 | 201.9 | 4.0 |
| | 20 | Prachuapkhiri-Khan Coast | 23 | 8.6 | 310.1 | 5.5 | 5 | 21.7 | 375.4 | 6.1 |
| | 14 | Mae Klong | 65 | 5.8 | 471.6 | 19.8 | 19 | 29.2 | 656.1 | 19.8 |
| South | 23 | Thale Sap Songkhla | 59 | 8.7 | 146.2 | 8.4 | 9 | 15.3 | 169.6 | 8.4 |
| | 21 | Peninsula-East Coast | 97 | 14.3 | 255.6 | 19.5 | 20 | 20.6 | 314.1 | 19.2 |
| | 25 | Peninsula-West Coast | 72 | 13.5 | 260.8 | 26.1 | 25 | 34.7 | 391.2 | 26.9 |
| | 24 | Pattani | 10 | 27.0 | 365.5 | 25.3 | 6 | 60.0 | 731.0 | 25.8 |
| | 22 | Tapi | 22 | 24.7 | 589.6 | 16.4 | 6 | 27.3 | 753.4 | 15.7 |
| Summation/Average | | | 1743 | 6.95 | 395.7 | 16.3 | 514 | 30.2 | 591.7 | 16.3 |

Note: St. is gauged rainfall station.

### 3.3. Pixel Based Comparison of SPE Products

As mentioned before, the monthly GGR data were compared with SPE products to determine the most accurate product for rainfall estimation in Thailand. For each product, the average RMSE and NSE between SPE and the GGR were calculated for all pixels. The spatial variation of RMSE and NSE are presented in Figure 4. To further simplify and visualise the variation in RMSE and NSE, an average value was taken for each region of Thailand and is presented in Table 4. CMORPH-BLD was the best performer, providing the highest average NSE of 0.800 and lowest RMSE of 49.6 mm. This was followed by TRMM-3B42, CHIRPS-PL, CHIRPS, and TRMM-3B42RT, respectively. Oceanic influences of the Andaman Sea and the Gulf of Thailand are likely to have contributed the reduced spatial correlations in southern Thailand.

### 3.4. Bias Correction of SPE Products

The NSE for each SPE product is presented in Table 5. Given the average NSE of 0.576 for all SPE products and the original GGR data, BC was applied in conjunction with DMCs in an attempt to increase the NSE. Improvements were evaluated by comparing the bias-corrected SPE and GGR datasets. This was iterated for all 180 sliding window periods and is presented as a line plot in Figure 5. As seen, the RABC, LBC, and DTBC methods provided very similar improvements, enhancing the average NSE between the GGR and SPE datasets to 0.801, 0.796, and 0.787, respectively. Therefore, the selection of the BC procedure was an insignificant factor in optimising the NSE. Instead, to effectively utilise the benefits of BC, it was found that the GGR data must first be reasonably accurate. By

taking the example of the RABC method, the application of the DMC procedure enhanced the average NSE by 14.9% from 0.576 to 0.662. As displayed in Figure 5b, this allowed for a total improvement of 39.1% to 0.801, which could not be attained by performing BC without first applying DMCs (i.e., GGR$_{Ori}$ dataset), wherein a 29.2% increase to 0.744 was observed.

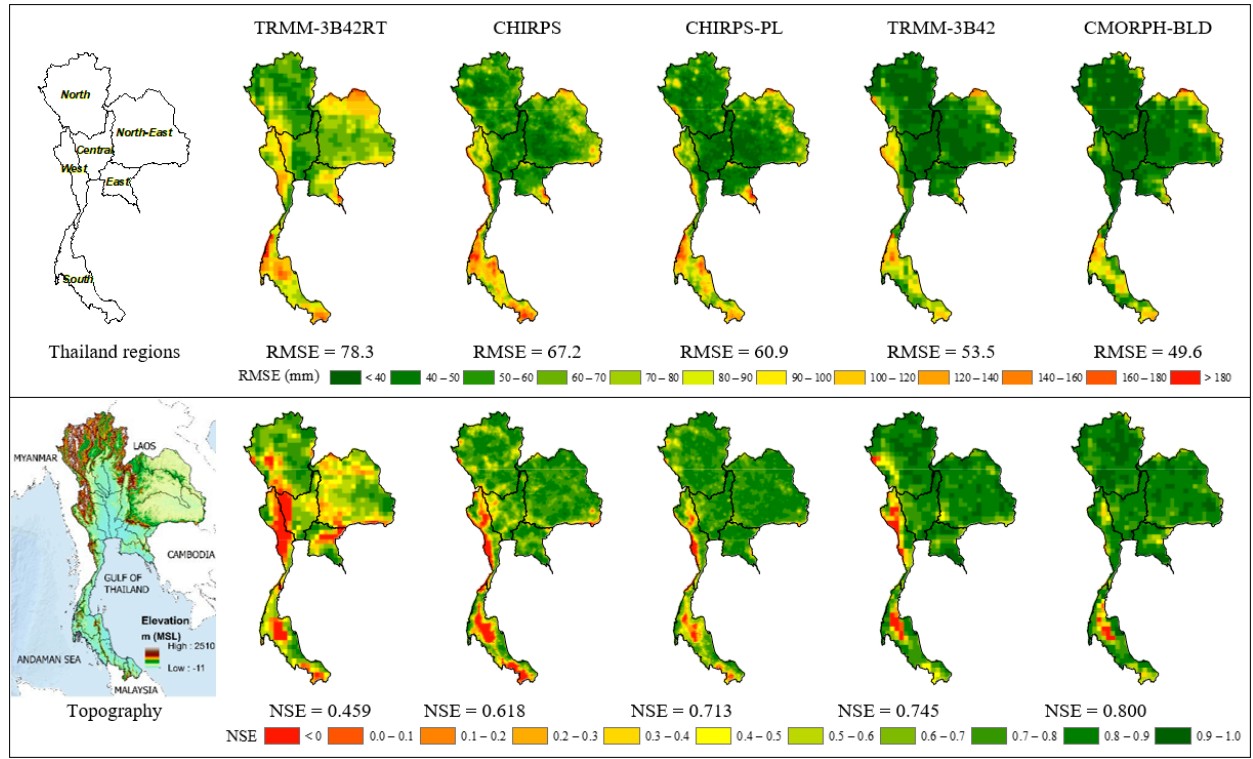

**Figure 4.** Spatial variation between GGR and SPE data from 2001 to 2015.

**Table 4.** Comparison between monthly average GGR and SPE rainfall estimates in the six regions of Thailand.

| Region | Indicator | TRMM-3B42RT | CHIRPS | CHIRPS-PL | TRMM-3B42 | CMORPH-BLD |
|---|---|---|---|---|---|---|
| Central | RMSE (mm) | 67.2 | 52.3 | 44.3 | 39.4 | **37.7** |
| | NSE | 0.393 | 0.655 | 0.760 | 0.797 | <u>0.817</u> |
| North | RMSE (mm) | 61.3 | 54.6 | 51.5 | 43.3 | **39.7** |
| | NSE | 0.621 | 0.720 | 0.761 | 0.795 | <u>0.852</u> |
| West | RMSE (mm) | 90.8 | 83.6 | 70.0 | 75.0 | **47.8** |
| | NSE | −0.028 | 0.151 | 0.435 | 0.366 | <u>0.749</u> |
| North-East | RMSE (mm) | 80.6 | 60.6 | 56.1 | **49.6** | 49.8 |
| | NSE | 0.525 | 0.746 | 0.790 | 0.824 | <u>0.829</u> |
| East | RMSE (mm) | 79.2 | 63.1 | 60.4 | **48.8** | 50.2 |
| | NSE | 0.488 | 0.736 | 0.780 | <u>0.820</u> | 0.806 |
| South | RMSE (mm) | 108.7 | 112.5 | 99.3 | 84.0 | **80.0** |
| | NSE | 0.304 | 0.299 | 0.525 | 0.603 | <u>0.634</u> |
| Thailand | RMSE (mm) | 78.3 | 67.2 | 60.9 | 53.5 | **49.6** |
| | NSE | 0.459 | 0.618 | 0.713 | 0.745 | <u>0.800</u> |

Note: Bolded values are the minimum RMSE for each region; underlined values are the maximum NSE for each region.

**Table 5.** Variation in NSE between SPE products and gauged rainfall datasets prior to and after undergoing DMC and BC procedures.

| SPE Product | Rainfall Dataset | | BC Method | Bias Correction (BC) | | | | NSE Improvements (%) | | |
| | Raw (GGR$_{ori}$) | DMC-Corrected (GGR) | | Calibration | | Validation | | Provided by BC to: | | Overall (DMC + BC) |
| | | | | GGR$_{ori}$ | GGR | GGR$_{ori}$ | GGR | GGR$_{ori}$ | GGR | |
|---|---|---|---|---|---|---|---|---|---|---|
| CMORPH-BLD | 0.710 | 0.792 (11.5%) | RABC | **0.812** | **0.862** | 0.782 | **0.843** | 10.1 | 6.5 | 18.7 |
| | | | LBC | 0.802 | 0.854 | **0.783** | 0.842 | 10.2 | 6.4 | 18.6 |
| | | | DTBC | 0.797 | 0.853 | 0.764 | 0.836 | 7.6 | 5.7 | 17.7 |
| TRMM- 3B42 | 0.639 | 0.728 (13.9%) | RABC | **0.811** | **0.859** | 0.780 | 0.839 | 22.1 | 15.2 | 31.3 |
| | | | LBC | 0.800 | 0.850 | **0.783** | **0.841** | 22.5 | 15.4 | 31.6 |
| | | | DTBC | 0.796 | 0.850 | 0.764 | 0.835 | 19.6 | 14.6 | 30.7 |
| CHIRPS | 0.525 | 0.622 (18.5%) | RABC | **0.768** | **0.815** | **0.736** | **0.793** | 40.2 | 27.6 | 51.2 |
| | | | LBC | 0.754 | 0.802 | 0.734 | 0.790 | 39.9 | 27 | 50.5 |
| | | | DTBC | 0.744 | 0.798 | 0.706 | 0.777 | 34.6 | 24.9 | 48 |
| CHIRPS-PL | 0.667 | 0.740 (10.9%) | RABC | **0.762** | **0.809** | **0.729** | **0.787** | 9.4 | 6.4 | 18 |
| | | | LBC | 0.742 | 0.790 | 0.719 | 0.776 | 7.8 | 4.9 | 16.3 |
| | | | DTBC | 0.737 | 0.791 | 0.698 | 0.770 | 4.7 | 4.1 | 15.4 |
| TRMM- 3B42RT | 0.337 | 0.430 (27.6%) | RABC | **0.728** | **0.767** | **0.690** | **0.740** | 104.7 | 72.3 | 119.6 |
| | | | LBC | 0.707 | 0.747 | 0.681 | 0.729 | 102 | 69.7 | 116.3 |
| | | | DTBC | 0.696 | 0.741 | 0.654 | 0.716 | 93.8 | 66.6 | 112.5 |
| Average | 0.576 | 0.662 (14.9%) | RABC | **0.776** | **0.822** | **0.744** | **0.801** | 29.2 | 20.9 | 39.1 |
| | | | LBC | 0.761 | 0.808 | 0.740 | 0.796 | 28.5 | 20.1 | 38.2 |
| | | | DTBC | 0.754 | 0.807 | 0.717 | 0.787 | 24.6 | 18.8 | 36.7 |

Note: Bolded values are the maximum NSEs among three BC methods; Underlined values are the maximum NSEs' percent of improvement among three BC methods.

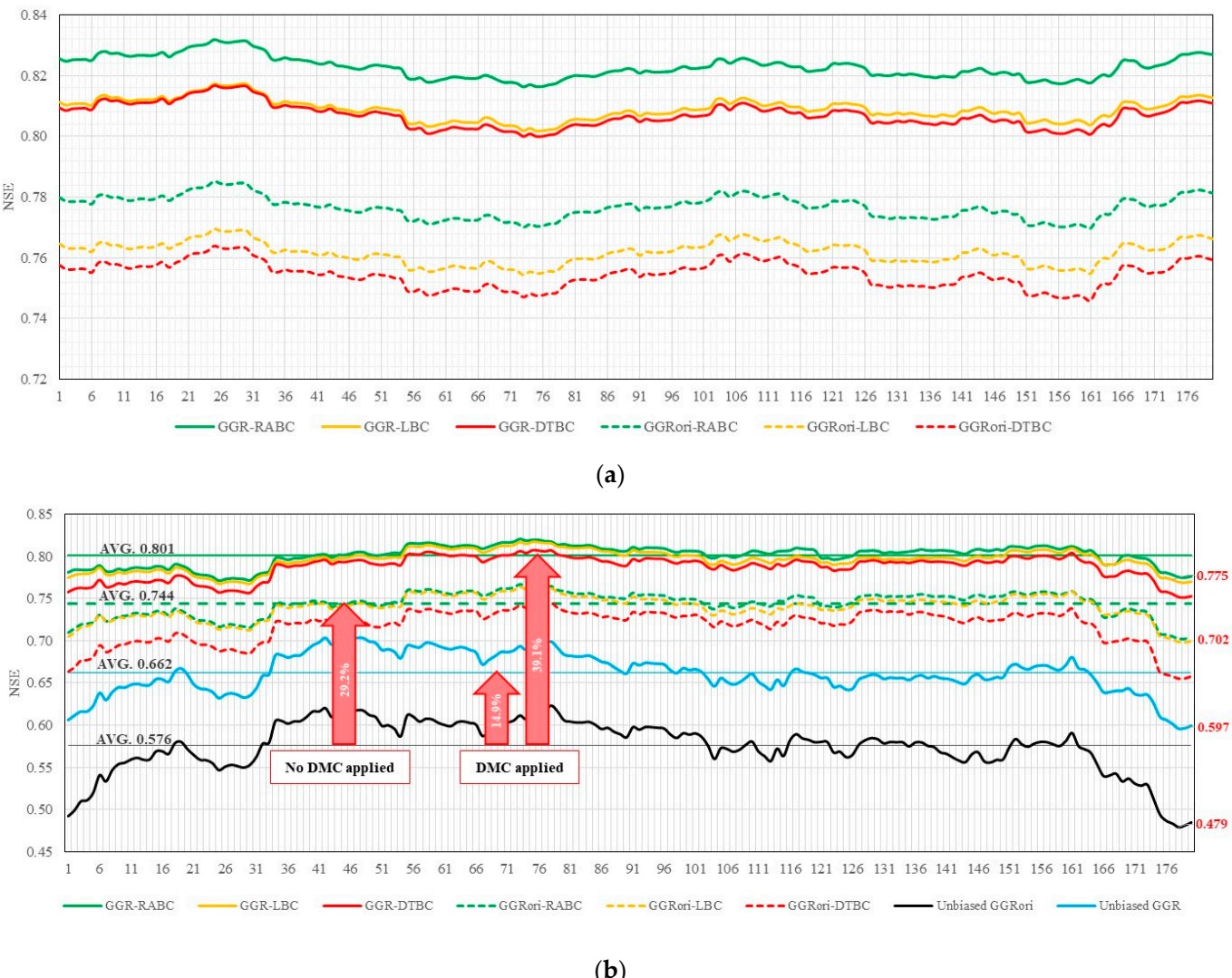

**Figure 5.** Validation of average NSE between gauged rainfall datasets prior to and after undergoing correction procedures for 180 sliding windows over the calibration and validation periods. (**a**) Calibration period; (**b**) validation period. Note: (1) AVG. 0.576 is the average NSE between SPE products and GGR$_{ori}$; (2) AVG. 0.662 is the average NSE between SPE products and GGR; (3) AVG. 0.744 is the average NSE between GGR$_{ori}$ and bias-corrected SPE products using RABC; (4) AVG. 0.801 is the average NSE between GGR and bias-corrected SPE products using RABC; and (5) the three arrows (labelled 14.9%, 29.2%, and 39.1%) denote the percentage improvement from (1) to (2), (1) to (3), and (1) to (4), respectively.

Inevitably, variations in GGR data will affect their degree of fit with SPE over the validation period—take, for instance, the poor NSEs of below 0.5 observed in the rearmost sliding windows in Figure 5b. Nevertheless, the impact of outlying data was effectively reduced, particularly with the combined usage of DMCs and BC (in this case, the RABC method), which lowered the range of NSE values from 0.479–0.623 (Δ = 0.144) to 0.775–0.821 (Δ = 0.046). The lone usage of BC, on the other hand, was able to limit this discrepancy to between 0.702 and 0.767 (Δ = 0.065).

Moreover, with the application of DMCs, the NSE of the GGR and SPE datasets became less sensitive to the selected BC procedure, particularly for the latter sliding windows where the discrepancy between the methods were noticeably marginalised. All in all, this clearly demonstrates the benefits of the BC procedures, which allow the corrected SPE datasets to be utilised with greater confidence.

It shall be noted that the influence of these BC procedures on individual SPE products is quite variable. Given that BC simply transforms the SPE dataset to produce a better fit with the gauged rainfall dataset, those with weaker original NSE values show notable changes. For instance, the NSE between GGR and CMORPH-BLD estimates improved by 19% from 0.710 to 0.843 after applying the RABC method with DMCs, whereas TRMM-3B42RT showed a 120% improvement. Amongst the five SPE products assessed in this study, the three best-performing ones were the post-real-time datasets, namely CMORPH-BLD, TRMM-3B42, and CHIRPS, which will subsequently be used to develop a weighted ensemble SPE product.

### 3.5. Ensemble Bias-Corrected SPE Products

Recommending a final SPE product for use might not help with obtaining optimal results for all locations and seasons, as the performance in an individual SPE product varies in time and space. It makes sense to use a SPE product at a given location and time of the year which provides the best rainfall match. Keeping this in mind, as the final part of the research, the three best-performing bias-corrected SPE (BSPE) products (Bias-CMORPH-BLD, Bias-TRMM-3B42, and Bias-CHIRPS) were pulled together to create an ensemble weighted average at each grid point. At a given location, the weights were allowed to vary with time and were formed on the basis of differences in SPE and GGR values following Equation (4). The product with the minimum error was allocated maximum weightage. These weights were then used to form the ensemble weighted average following Equation (5).

$$W_i = \frac{1/\left(BSPE_i - GGR\right)^2}{\sum_{j=1}^{n} 1/\left(BSPE_j - GGR\right)^2} \tag{4}$$

$$SPE_E = \sum_{i=1}^{n} W_i BSPE_i \tag{5}$$

In order to access the fidelity of the proposed approach, a cross validation in space was adopted. The procedure involved randomly dropping 30% of the grid points, calibrating the model at the remaining 70% of locations, and noting down the weights estimated using Equation (5). These weights were then used to simulate rainfall time series at the dropped 30% locations and calculated the NSE using the observed and simulated time series. The whole procedure was repeated 20 times to obtain the unbiased results.

The weighted averaged results so obtained are presented in Figure 6 in the form of bar charts of NSE for both the calibration and validation periods. For comparison, NSEs of individual BSPE products using RABC are also included in the figure. The NSEs for calibration were obtained by comparing the GGR time series at 70% of the locations with the BSPE. For validation, NSEs were compared between the GGR time series at the remaining 30% of grid points and the BSPE from the surrounding pixels. These procedures were also repeated 20 times. As can be seen from the results presented in Figure 6, the combined product provides an improved NSE in comparison to each individual BSPE products irrespective of the calibration and validation periods.

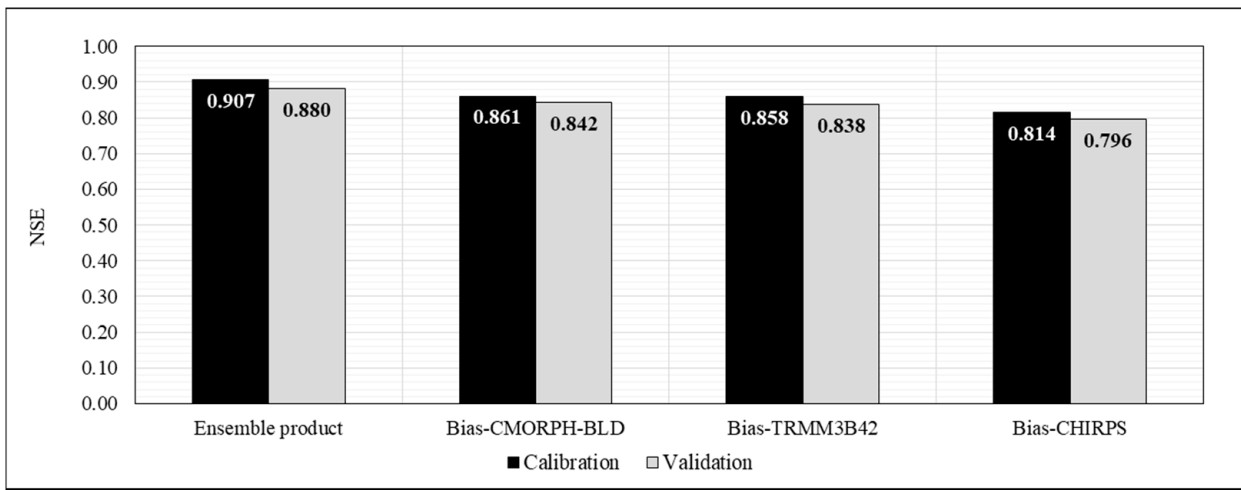

**Figure 6.** Comparison of the accuracy between the ensemble BSPE products and individual BSPE products.

Figure 7 provides the spatial variation of averaged NSEs across 20 simulations. Overall, irrespective of the regions or time periods, the approach provides high NSEs over the country, barring a few grid points in the south during validation. These results are encouraging and suggest that after BC and forming a weighted average, these BSPE products can be used in data-scarce locations with confidence.

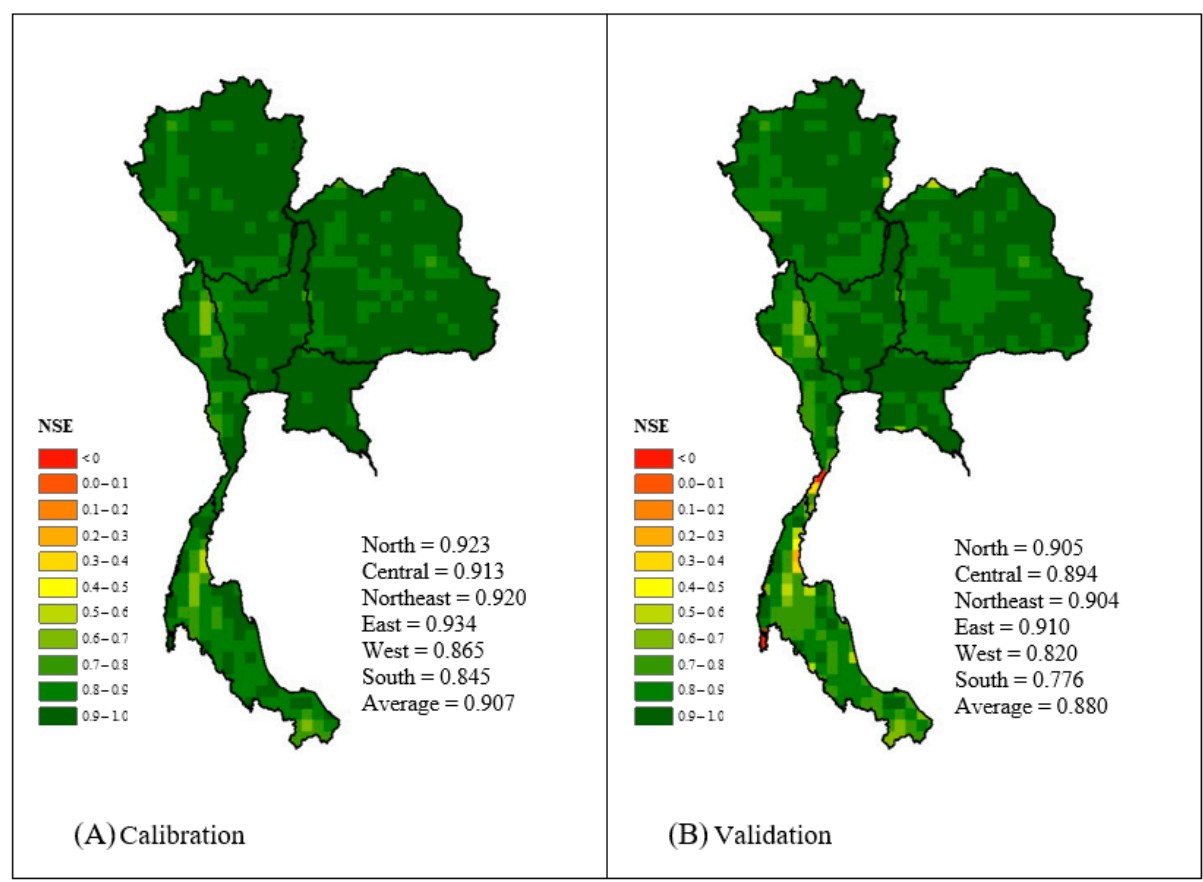

**Figure 7.** Spatial cross-validation of average NSE between GGR and weighted ensemble SPE rainfall product.

## 4. Conclusions

The importance of acquiring good-quality rainfall data is ever-increasing. The ease of access of SPE products has made them a demanding area of research. Yet, the biases in the SPE products and continual quest to produce reliable areal rainfall estimates still limits SPE from being confidently applied for water resource management purposes. This study demonstrates the effectiveness of validating monthly gauged rainfall data prior to their use in SPE validation. By using conventional DMC to eliminate questionable gauged rainfall data, this led to the elimination of around 7% of gauged rainfall data and reduced the RMSE between the station of interest and its surroundings by 16.3 mm. The improved accuracy, which was equivalent to around a 33% increase in the density of the observation network (from 592 to 396 km$^2$ per station), further validates the usefulness of DMCs.

The BSPE datasets without DMC provided significant improvements to the NSE (25–29%). Further, 10% (total 36–39%) improvements in NSE were gained by combining the DMC and BC. As a final part of the research, the final DMC and BSPE products were pooled together to form the weighted ensemble average. The results show that the combined product further improves rainfall estimates at left-out locations.

The research emphasises the need to check the quality of gauged rainfall data prior to performing BC in order to effectively enhance the accuracy of the SPE products. The weighted averaged SPE product further enhances the rainfall estimates and highlights the usefulness of the approach in obtaining rainfall in inaccessible and data-sparse regions.

**Author Contributions:** Conceptualization, N.S.; methodology, N.S. and C.K.; software, C.K. and K.T.; validation, C.K. and K.T.; formal analysis, C.K.; investigation, C.K. and N.P.; data curation, N.P. and K.T.; writing—original draft preparation, N.S. and C.K.; writing—review and editing, J.A.W.; visualization, N.S. and J.A.W.; supervision, W.G.M.B. and W.S.; funding acquisition, N.S. All authors have read and agreed to the published version of the manuscript.

**Funding:** This research was funded by: (1) Remote Sensing Research Centre for Water Resources Management (SENSWAT), Faculty of Engineering, Kasetsart University, under the project number 62/03/WE., and (2) Master's Degree Research Assistantship Program, Faculty of Engineering, Kasetsart University, under the project number 64/03/WE/M.Eng.

**Data Availability Statement:** Data are available from the corresponding author by request.

**Acknowledgments:** We gratefully acknowledge the Royal Irrigation Department and Thai Meteorology Department for providing the daily rainfall data. We also truthfully thank the Climate Hazards Group, National Aeronautics and Space Administration (NASA), Center for Hydrometeorology and Remote Sensing (CHRS), and National Oceanic and Atmospheric Administration (NOAA) for creating and sharing the SPE data to be used in this study.

**Conflicts of Interest:** The authors declare no conflict of interest.

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
