# Peer review of "The Combined Power of Double Mass Curves and Bias Correction for the Maximisation of the Accuracy of an Ensemble Satellite-Based Precipitation Estimate Product"

_hydrology, doi:10.3390/hydrology10070154_

Round 1
Reviewer 1 Report
GENERAL COMMENTS
The manuscript is a step forward in using satellite based tropical precipitation estimates corrected with existing gauge networks.
COMMENTS ON DETAILS
1. Introduction.
a. "Poor spatial resolution" is mentioned, but it would be interesting to express satellite precipitation estimate resolutions more comparable to the gauge density figures, what would be the area of a 0.25 or 0.05 degrees resolution. Also what is the degrees in the resolution, I suppose latitude or longitude grid related not to the satellite swath?
2. Materials and methods.
a. Figure 1 could have background with light colours to make the points of gauges better visible.
b. LIS is probably not for "lighting" but "lightning"!
c. In "Usefulness of DMC" the sentence "Thus,... compared to original data was removed..." may be missing something or has too much?
d. In "Cross-validation of bias correction procedures" the last sentence "were determine" may need correction.
3. Results and discussion.
a. In table 2 some figures in bold, and in table 4 bold and underlined, but I see no clear explanation why.
b. You could comment on the obvious differences in spatial correlations between various areas of Thailand. Especially the land area between large bodies of water catches the eye.
Author Response
Answer to Reviewer#1
Comments and Suggestions for Authors GENERAL COMMENTS
The manuscript is a step forward in using satellite based tropical precipitation estimates corrected with existing gauge networks.
COMMENTS ON DETAILS
- Introduction.
- "Poor spatial resolution" is mentioned, but it would be interesting to express satellite precipitation estimate resolutions more comparable to the gauge density figures, what would be the area of a 0.25 or 0.05 degrees resolution. Also what is the degrees in the resolution,
I suppose latitude or longitude grid related not to the satellite swath?
Answer:
With regards to second sentence, resolution is commonly expressed in the degrees format, since the spatial coverage of the satellite swath (e.g. 50oS to 50oN for TRMM-3B42 V7) represents different values in metric units depending on latitude. For instance, 0.00045º is about 50m around the Equator, but about 30m around London. As such, we shall express the resolution in this manner as per the norm.
Further, for this reason, in our area of study, the spatial coverage of 0.25o and 0.05o is approx. 770 km2 and 30 km2, respectively – which would significantly differ elsewhere.
These numbers have been mentioned in the revised paragraph, making it clearer for the sake of comparison with the density of gauged rainfall data as suggested.
- Materials and methods.
- Figure 1 could have background with light colours to make the points of gauges better visible.
Answer: The background in Figure 1 has been changed accordingly.
- LIS is probably not for "lighting" but "lightning"!
Answer: The term "lighting" has been changed to "lightning" in the revised manuscript.
- In "Usefulness of DMC" the sentence "Thus,... compared to original data was removed..." may be missing something or has too much?
Answer: The sentence has been modified to better convey the applied approach.
From: Thus, to alternatively recreate the same effect, stations were randomly removed from the gauged rainfall dataset to induce an increase in the root mean square error (RMSE) in the areal rainfall estimates compared to original data was removed from the validated gauged rainfall dataset (result of section 2.2.1).
To: Hence, as a means to replicate the outcome from Section 2.2.1, stations were randomly removed from the gauged rainfall dataset to induce an increase in the root mean square error (RMSE) in the areal rainfall estimates relative to utilising the full dataset.
- In "Cross-validation of bias correction procedures" the last sentence "were determine" may need correction.
Answer:
The sentence was replaced with “The average NSE value from 180 iterations of sliding windows for each of SPE products were determined to assess the overall performance.”
- Results and discussion.
- In table 2 some figures in bold, and in table 4 bold and underlined, but I see no clear explanation why.
Answer:
Beneath Table 2, we have explained the use of bold numbers, as well as the highlighting of the row corresponding to the suitable NSE threshold as below:
Rows with bolded numbers corresponds to the NSE thresholds which yields the maximum NSE after dropping the data/stations for each SPE product; The highlighted row is the suitable NSE threshold used to produce the validated gauged rainfall dataset.
Similarly, for Table 4, the following sentenced was added:
Bolded value is the minimum RMSE for each region; Underlined value is the maximum NSE for each region.
- You could comment on the obvious differences in spatial correlations between various areas of Thailand. Especially the land area between large bodies of water catches the eye.
Answer:
Oceanic influences of the Andaman Sea and the Gulf of Thailand are likely to have contributed the reduced spatial correlations in southern Thailand. This statement was added as the last sentence in section 3.3 (Pixel based comparison of SPE products).

Reviewer 2 Report
Review report of the manuscript:
The combined power of double mass curves and bias correction for the maximization of the accuracy of an ensemble satellite-based precipitation estimate product
Summary:
The study addresses the challenge of accurately estimating rainfall characteristics by improving the reliability of gauged rainfall data and enhancing the accuracy of satellite-based precipitation estimates (SPE) in Thailand. By validating five SPE products with 1,779 gauged rainfall stations, the researchers employed a double mass curve (DMC) analysis to eliminate inconsistent readings and improve the Nash-Sutcliffe Efficiency (NSE) by 13.9%. They also applied three bias correction (BC) procedures to the SPE products using gauge-based gridded rainfall (GGR). Combining the DMC and BC procedures significantly improved the performance of the SPE products. Finally, by using an ensemble weighted average of the three best-performing bias-corrected SPE products, the NSE reached 0.907 during calibration and 0.880 during validation, providing an accurate method for obtaining daily rainfall data in remote locations.
General concept comments:
The manuscript is clear, relevant to the field, and presented in a well-structured manner. I found the applied methodology appropriate for achieving the main objective of the study, which is to improve the accuracy of several SPE products covering Thailand river basins by combining them to form a weighted average series.
i) However, it is unfortunate that the testing period is quite limited due to the availability of the SPE products.
ii) Some of the figures and tables require better descriptions to enhance the comprehension of potential readers. There are issues such as describing the meaning of the bold values in Tables 2, 4, and 5, highlighting specific rows in Table 2, and clarifying the meaning of the arrows in Figure 5.
It seems to me that the manuscript is well written and requires only minor editing of the English language.
Author Response
Answer to Reviewer#2
Comments and Suggestions for Authors
Review report of the manuscript:
The combined power of double mass curves and bias correction for the maximization of the accuracy of an ensemble satellite-based precipitation estimate product
Summary:
The study addresses the challenge of accurately estimating rainfall characteristics by improving the reliability of gauged rainfall data and enhancing the accuracy of satellite-based precipitation estimates (SPE) in Thailand. By validating five SPE products with 1,779 gauged rainfall stations, the researchers employed a double mass curve (DMC) analysis to eliminate inconsistent readings and improve the Nash-Sutcliffe Efficiency (NSE) by 13.9%. They also applied three bias correction (BC) procedures to the SPE products using gauge-based gridded rainfall (GGR). Combining the DMC and BC procedures significantly improved the performance of the SPE products. Finally, by using an ensemble weighted average of the three best-performing bias-corrected SPE products, the NSE reached 0.907 during calibration and 0.880 during validation, providing an accurate method for obtaining daily rainfall data in remote locations.
General concept comments:
The manuscript is clear, relevant to the field, and presented in a well-structured manner.
I found the applied methodology appropriate for achieving the main objective of the study, which is to improve the accuracy of several SPE products covering Thailand river basins by combining them to form a weighted average series.
Answer:
The summary above is well-aligned with the key takeaways we hope readers will gain from this manuscript.
- i) However, it is unfortunate that the testing period is quite limited due to the availability of the SPE products.
Answer:
The acquisition of SPE datasets was indeed a challenging task, particularly given the spatial extent of this study (nationwide). Had it been a more straightforward process, we would have undoubtedly ensured a longer testing period for examining our methodology.
- ii) Some of the figures and tables require better descriptions to enhance the comprehension of potential readers. There are issues such as describing the meaning of the bold values in Tables 2, 4, and 5, highlighting specific rows in Table 2, and clarifying the meaning of the arrows in Figure 5.
Answer:
We appreciate you pointing this out to us. Descriptions have been added to each of the tables and to Figure 5 for better clarity.
Comments on the Quality of English Language
It seems to me that the manuscript is well written and requires only minor editing of the English language.
Answer:
We have made the changes as per both reviewers’ suggestions, which was also the opportunity for us to revise our manuscript overall. Thank you for your feedback.
